# Magnetic Resonance Imaging Features of Anterolateral Ligament in Young Adults without Anterior Cruciate Ligament Injury: Preliminary Evaluation

**DOI:** 10.3390/diagnostics14121226

**Published:** 2024-06-11

**Authors:** Ji-Hee Kang, Sung-Gyu Moon, Dhong-Won Lee

**Affiliations:** 1Department of Radiology, Konkuk University Medical Center, Seoul 05030, Republic of Korea; 20200184@kuh.ac.kr; 2Department of Orthopedic Surgery, Konkuk University Medical Center, Seoul 05030, Republic of Korea; osdoctorknee@kuh.ac.kr

**Keywords:** anterolateral ligament, anatomy, knee, magnetic resonance imaging

## Abstract

This study aimed to characterize the Magnetic Resonance Imaging (MRI) features of the Anterolateral Ligament (ALL) in young adults without Anterior Cruciate Ligament (ACL) injury and evaluate its visibility using MRI. In this retrospective analysis, MRI scans of 66 young adults without ACL injuries were assessed by two radiologists. The ALL was examined from its bone-to-bone attachment between the lateral femoral epicondyle and the lateral tibia. The visibility of the ALL was classified as normal, probably normal, abnormal, or non-visualized, based on ligament continuity and thickness relative to the Meniscotibial Ligament (MTL). A continuous structure with thickness equal to or greater than the MTL was considered normal; continuous but wavy and thin features were categorized as probably normal; discontinuity and angulation were deemed abnormal. The proximal attachment of the ALL was categorized as anterior, central, or posterior to the Fibular Collateral Ligament (FCL), while the distal attachment was noted as either at the same location or distal to the MTL. The ALL was identified in 87.9–95.5% of knees and was non-visualized in 4.5–12.1% of cases. Continuous ligamentous structures were observed in 63.7–71.2% of knees (normal in 30.3–37.9%; probably normal in 27.3–40.9%), whereas 19.7–30.3% exhibited abnormal features. Inter-observer agreement was moderate to substantial (κ = 0.66, 0.56), and intra-observer agreement was substantial to excellent (κ = 0.82, 0.66). Among the 58 visible ALLs, proximal attachments were predominantly anterior (63.8%) or central (32.8%) to the FCL, with a minority posterior (1.7%). In total, 4 of the 19 central insertions were incorporated into the FCL mid-substance, and one case was blended into the meniscofemoral ligament. Distal attachments were equally distributed between the same location (50%) and distal to the MTL (50%) (mean 3.7 mm distal). In conclusion, MRI was feasible for detecting the ALL in most young adults without ACL injury, revealing continuous ligament structures in about two-thirds of cases. Approximately 40% of cases exhibited a thickness equal to or greater than the MTL, with the majority of proximal attachments located anterior to the FCL and distal attachments evenly divided between the same insertion and distal to the MTL.

## 1. Introduction

Since the initial reports on the anterolateral ligament (ALL) by Vincent et al. in 2012 [1], interest in the ALL has increased due to its potential impact on knee stability and its association with anterior cruciate ligament (ACL) injuries [2,3,4]. The existence of a distinct ligament within the anterolateral capsular complex remains controversial [5,6,7,8,9,10]. Urban et al. suggested that the ALL is an artificial construct formed by the anterolateral part of the iliotibial tract and the aponeurotic tibial insertion of the biceps femoris tendon [11]. Shea et al. reported varying incidences of ALL in children, ranging from 14% to 64% [6]. However, several anatomical studies [2,12] have identified the ALL as a distinct structure with histological characteristics of ligaments, observed in 100% of cadaveric fetuses [13] and approximately 82.9% of adult dissections [14]. Variations in detection rates are likely due to differences in dissection methodologies [5,8].

Anatomic studies have described the ALL as having attachment points connecting the lateral femoral epicondyle to the lateral tibia [12]. Dissection studies have identified several common characteristics, including an extracapsular structure following an oblique course, with the femoral attachment near the lateral epicondyle, and the tibial attachment at the anterolateral aspect of the tibia between Gerdy’s tubercle and the fibular head [13]. Several authors reported that the ALL attaches distally to the lateral meniscus as well as the tibial plateau [2,12], although this remains controversial. Consequently, anatomical descriptions vary, reflecting ongoing development in our understanding of this structure.

Magnetic Resonance Imaging (MRI) is a non-invasive and detailed method for evaluating the appearance and characteristics of knee ligaments [15,16,17,18]. MRI has long been used as an effective tool for visualizing ALL, with a focus on the coronal plane and T2-weighted images. High-resolution three-dimensional pulse sequences also provide comprehensive evaluations of the ALL [18]. A systemic review of MRI studies found that ALL was visualized in 84.80% of cases [19], a detection rate comparable to adult dissections [14]. Previous studies have employed MRI to assess the ALL primarily in cadaver knees or in the context of ACL injuries. However, there is a notable lack of research on the MRI characteristics of the ALL in the absence of ACL injury [18,20]. Gaining insights into the MRI features of the ALL without ACL injury could enhance knowledge of its potential role, aid in identifying ALL-related pathologies, and potentially contribute to treatment approaches.

We hypothesized that in subjects without ACL injury, where the ALL is expected to be intact, the ALL would have a unique appearance on MRI and distinct proximal and distal attachment sites compared to other fascia or retinaculum structures. The purpose of this study is to describe the MRI imaging characteristics of ALL in young adults without ACL injury, focusing on its morphology, continuity, and attachment points. Additionally, we aim to assess the visibility of the ALL using routine 3-Tesla MRI.

## 2. Materials and Methods

### 2.1. Study Population

This retrospective study received approval from our hospital’s institutional review board, which waived the need for informed consent due to the study’s retrospective design. This study was conducted following the STROBE (Strengthening the Reporting of Observational Studies in Epidemiology) Guidelines and complied with the ethical standards established by the 1964 Declaration of Helsinki and its subsequent amendments.

Between January and December 2017, our hospital conducted 973 knee MRI scans. Of these, 115 met the inclusion criteria: (a) 3-T MRI study with a standardized protocol and (b) patients aged 16 to 29 years. We excluded 49 scans due to prior ACL or lateral compartment capsule surgery (30 cases), confirmed ACL injuries via arthroscopy (15 cases), clinical diagnoses of ACL injury with pivot-shift bone marrow edema (2 cases), and lateral compartment fractures (2 cases). Consequently, 66 knee MRI scans were included in the study (Figure 1). Electronic medical records and surgical notes were reviewed for each case, documenting age, gender, surgical history, and follow-up period.

### 2.2. MRI Protocol

MR examinations were performed using two 3-Tesla scanners: a Magnetom Skyra (Siemens Healthcare Diagnostics, Erlangen, Germany) using a 16-channel phased array coil (*n* = 44) and a Discovery MR750 (GE Healthcare, Milwaukee, WI, USA) with a 16-channel coil (*n* = 22). The 2D imaging sequences are as follows: for Magnetom Skyra, sagittal T1-weighted (TR/TE, 695/11; number of excitations [NEX], 2; matrix, 512 × 282; thickness, 3 mm; field of view [FOV], 16 cm), sagittal T2-weighted (4050/76; 2; 512 × 282; 3 mm; 16 cm), coronal T2-weighted (3900/76; 2; 512 × 281; 3 mm; 16 cm), oblique coronal T2-weighted (3400/68; 4; 384 × 211; 3 mm; 14 cm), and axial T2-weighted fat-suppressed (4130/58; 3; 384 × 269; 3 mm; 15 cm). For Discovery MR750, sagittal T1-weighted (TR/TE, 840/17; NEX, 1; matrix, 512 × 256; thickness, 3 mm; FOV, 16 cm), sagittal T2-weighted (3750/73; 1; 640 × 320; 3 mm; 16 cm), coronal T2-weighted (3750/73; 1; 640 × 320; 3 mm; 16 cm), oblique coronal T2-weighted (2500/71; 2; 416 × 256; 3 mm; 14 cm), and axial T2-weighted fat-suppressed (3500/64; 2; 352 × 288; 3 mm; 15 cm). There was also a 3D-isotropic sagittal proton-density-weighted fat-suppressed (900/27; 1; 320 × 320; 1 mm; 16 cm for Magnetom Skyra, 1302/36; 1; 320 × 320; 0.8 mm; 16 cm for Discovery MR750) image with coronal, axial reconstruction image without interslice gap. The detailed image parameters are presented in Table 1.

### 2.3. Image Analysis

Two musculoskeletal radiologists (J.H.K. and S.G.M., with 4 and 18 years of experience, respectively) independently reviewed the MRI images. They were blinded to the clinical information and diagnoses of the subjects. The visibility of the anterolateral ligament (ALL) was evaluated using coronal T2-weighted and oblique coronal T2-weighted imaging planes on a picture archiving and communications system (PACS). Fat-suppressed 3D-isotropic proton-density-weighted images were excluded from image analysis due to the inability to distinguish detailed anatomy of thin and small structures because of blur effect and low contrast-to-noise ratio. Measurements were taken using the measurement tools of Centricity PACS 6.0 (GE Healthcare, Mt. Prospect, IL, USA).

Schematic illustrations of the courses of the ALL are shown in Figure 2. Each reader independently assessed the morphology (continuity, waviness, thickness, and other features), proximal femoral attachment site, and distal tibial attachment site. The proximal femoral attachment site was classified as anterior, center, posterior to the fibular collateral ligament (FCL), or other (e.g., mid-substance of the FCL, blended with the meniscofemoral ligament). The distal tibial attachment sites were classified as the same insertion or distal insertion to the meniscotibial capsular ligament (MTL). Axial T2-weighted fat-suppressed images were used as a reference to determine the positional relationship of the ALL to the FCL ligament or MTL ligament. In cases of distal insertion, the distance from the MTL attachment site was measured using the most internal fibers of the ligament as a reference in millimeters. If the insertion sites of the MTL and ALL were not in the same coronal plane, the perpendicular distance between the insertion points of the MTL and ALL was measured with the help of axial T2-weighted fat-suppressed images. The thickness of the ALL was evaluated as thinner, equal, or thicker in comparison to that of the MTL.

Based on ALL morphology, the visibility of the ALL was graded as normal, probably normal, abnormal, or invisible. “Normal” was assigned when the entire length of the ALL appeared straight or convex, had even thickness, and was thicker than or equal to the MTL. “Probably normal” indicated that the ALL was continuous but had a wavy or thinner appearance compared to the MTL. “Abnormal” referred to the presence of discontinuity, angulation, or partial irregularity. “Invisible” indicated the absence of a clearly identified ligament along the course and indistinct femoral or tibial insertion.

Due to the large number of small anatomical components of the lateral complex of the knee (e.g., anterolateral ligament, lateral collateral ligament, iliotibial band, popliteal tendon, biceps femoris tendon, and meniscocapsular ligaments), both readers were cautious while evaluating the cases to avoid potential errors such as the partial-volume effect or misinterpretation of structures. Therefore, to prevent over-classification of the ALL as visible, the readers only considered it present when clearly visible in both sequences, with cross-referencing of the images (coronal T2-weighted and oblique coronal T2-weighted images). The ALL was classified as invisible if it only appeared in one sequence. To assess intra-observer variability, both readers re-evaluated the images after an 8-week interval, without knowledge of the previous analysis results.

### 2.4. Statistical Analyses

Inter-observer and intra-observer agreements were calculated to determine the visualization of ALL using weighted kappa statistics. Kappa values were interpreted as follows: 0.01–0.20, slight; 0.21–0.40, fair; 0.41–0.60, moderate; 0.61–0.80, substantial; and 0.81–1.00, excellent. Data analysis for the morphology of ALL was performed by the more experienced reader (Reader 2); Reader 1’s interpretation was used to assess inter-observer agreement.

The correlation between ALL visualization and previous surgery, knee side, age, and gender was analyzed using the Spearman correlation coefficient. Power analysis for the paired sample t-tests, setting an alpha error at 0.05 and aiming for a power of 0.8, determined that a minimum sample size of 29 knees was necessary.

Statistics were calculated using SPSS™ version 25 (IBM Corp., Armonk, NY, USA) and MedCalc version 16.2.1 (MedCalc Software, Ostend, Belgium), while power analysis was conducted using G*Power software 3.1.9.4 (Heinrich-Heine-Universität Düsseldorf, Düsseldorf, Germany). A *p*-value of less than 0.05 was considered statistically significant. 

## 3. Results

### 3.1. Patient Characteristics

The study group consisted of 66 knee MRI scans from 64 subjects (2 subjects had both knees scanned). Among the 64 patients, 13 were female and 51 were male, with an average age was 22.8 years (range, 16–29 years). The scans included 31 right knees and 35 left knees, with a mean follow-up period of 16.5 weeks (range, 7 days–47 months). Sixteen knees had a history of prior arthroscopic surgery unrelated to the ACL or lateral compartment capsule: partial meniscectomy of the lateral (*n* = 9) or medial meniscus (*n* = 3), PCL reconstruction (*n* = 3), and microfracture of osteochondritis dissecans (*n* = 1). In 38 cases, an intact ACL was confirmed via arthroscopy. Postoperative diagnosis included lateral meniscus tear (*n* = 12), PCL tear (*n* = 7), medial meniscus tear (*n* = 5), cartilage erosion (*n* = 4), osteochondritis dissecans (*n* = 3), patella dislocation (*n* = 3), quadriceps tendinopathy (*n* = 1), medial patella plica (*n* = 1), Osgood–Schlatter disease (*n* = 1), and ganglion cyst (*n* = 1). In the remaining 28 cases, an intact ACL was clinically diagnosed based on medical history and physical examinations, including anterior drawer and pivot-shift tests. These cases included normal MRI findings (*n* = 11), soft tissue contusion (*n* = 3), cartilage erosion (*n* = 2), PCL partial tear (*n* = 2), patella tendinopathy (*n* = 2), synovitis (*n* = 2), ganglion (*n* = 1), osteochondritis dissecans (*n* = 1), thigh muscle sarcoma (*n* = 1), medial meniscus partial tear (*n* = 1), effusion (*n* = 1), and medial patella plica (*n* = 1).

### 3.2. ALL Visibility

The visibility of the ALL, as assessed by two readers, is summarized in Table 2. Reader 1 identified the ALL in 90.9–95.5% of knees: normal (30.3–36.4%), probably normal (30.3–40.9%), and abnormal (19.7–28.8%). The ALL was not visualized in 4.6–9.1% of cases. Reader 2 found the ALL in 87.9–95.5% of knees: normal (36.4–37.9%), probably normal (27.3%), and abnormal (24.2–30.3%). The ALL was not visualized in 4.6–9.1% of cases. Overall, the ALL was identified in 87.9–95.5% of knees, with 63.7–71.2% classified as normal or probably normal, showing continuous ligamentous features, and 19.7–30.3% as abnormal, showing discontinuity or angulation. The ALL was invisible in 4.5–12.1% of cases.

The inter-observer agreement was moderate to substantial (κ = 0.66, 0.56), and the intra-observer agreement was substantial to excellent (κ = 0.82, 0.66). There was no correlation between ALL visibility grade and previous surgery (*p* = 0.15), knee side (*p* = 0.26), age (*p* = 0.58), gender (*p* = 0.48), or MRI unit (*p* = 0.71).

### 3.3. ALL Morphology

As summarized in Table 3, Reader 2’s interpretation of the 58 visible cases revealed the proximal femoral attachment sites: anterior to the FCL (*n* = 37, 63.8%), central to the FCL (*n* = 19, 32.8%), posterior to the FCL (*n* = 1, 1.7%), and blended into the meniscofemoral ligament (*n* = 1, 1.7%). Of the 19 central insertions, 4 were attached to the mid-substance of the FCL and consequently incorporated into the FCL (*n* = 4, 6.9%). The distal tibial attachment sites were either the same as the MTL (*n* = 29, 50%) or distal to the MTL (*n* = 29, 50%), with a mean distance from the ALL to the MTL of 3.7 mm (range, 1.7–6.1 mm). ALL thicknesses were thicker (*n* = 10, 17.2%), equal (*n* = 26, 44.8%), or thinner (*n* = 22, 37.9%) compared to the MTL. Overall, 63.7% of cases showed continuity, while 36.3% showed discontinuity. Other features included ligamentous duplication (*n* = 6, 9.1%) and intrasubstance splitting (*n* = 2, 3.0%), with similar attachment sites at the proximal and distal ends. Representative cases are shown in Figure 3, Figure 4, Figure 5, Figure 6 and Figure 7.

## 4. Discussion

The main results of this study show that in approximately 90% of young adults without ACL injury, the ALL was detectable on MRI. In about two-thirds of the cases, MRI revealed continuous ligament structures. Additionally, around 40% of these structures were found to be either thicker than or similar in size to the MTL. The majority of the proximal attachment was found to be located anterior to the FCL, while the distal attachment was found to be either at the same insertion point or distal to the MTL with equal frequency.

The ALL potentially contributes to rotatory stability as a synergist with the ACL. Detailed anatomical dissection studies have provided descriptions of the ALL’s presence, pathway, and attachment [1,12]. They confirmed the presence of the ALL in all dissected knees and described it as an obliquely running ligament with a consistent origin and insertion site. The ligament originates from the lateral femoral epicondyle and attaches to the anterolateral aspect of the proximal tibia. Caterine et al. provided additional information on anatomical variations in the ligament’s origin and insertion through anatomical dissection and histological analysis [21]. The study demonstrated that ALL is a clearly defined ligamentous structure that is separate from the joint capsule. Histological analysis also confirmed the presence of connective tissue that corresponds to a typical ligament.

ALL injuries are commonly associated with ACL injuries, with MRI reports showing prevalence rates as high as 79% in ACL injured knees [22]. Recent reports indicate that 19.2% to 29.3% of patients with a complete ACL tear also have a concomitant ALL tear on MRI [23,24]. Many institutions have adopted additional ALL reconstructions for ACL injuries with rotational instability. Combined ACL and ALL reconstruction has demonstrated greater mid-term rotational stability and significantly lower re-rupture rates compared to isolated ACL reconstruction [25]. In order to assist with graft placement during ALL reconstruction, Rezansoff et al. have described the radiographic landmarks of the ALL origin and insertion [26]. In addition, fluoroscopy was used to demonstrate the specific attachment points of ALL. In one anatomical variant, the femoral origins of ALL were found to be 3.3 +/− 1.5 mm anterior and distal to the FCL origin, while in the second variant, they were 5.4 +/− 1.4 mm posterior and proximal to the FCL origin. On average, the tibial insertions of ALL were located 24.7 +/− 4.5 mm posterior to Gerdy tubercle and 11.5 +/− 2.9 mm distal to the lateral tibial plateau [26].

Studies have utilized MRI to examine the properties of the ALL. Ariel et al. [19], in a systematic review of anatomical studies, reported that the ALL was observed in 84.8% of overall MRI studies. However, the detection rates in dissection studies varied significantly, ranging from 4% to 100% [9,27]. These discrepancies between MRI and cadaveric dissection detection rates may be attributed to technical variations. Evaluating the anatomical structure of the ALL based solely on coronal MRI images, as conducted by Claes et al. [22] and Helito et al. [28], may lead to potential anatomical pitfalls and partial-volume effects, particularly given the ALL’s thin, short structure and its close intertwinement with the surrounding fibers of the lateral ligamentous complex of the knee [14]. In contrast, our study employed specific criteria for assessing the visibility of the ALL using both coronal and oblique coronal sequences, resulting in MRI detection rate for the ALL between 87.9% and 95.5%.

The majority of existing studies have focused on evaluating the ALL in cadaveric knees or in the context of ACL injuries. There is notable paucity of research examining the MRI characteristics of the ALL in the absence of ACL injury [18,20]. In studies involving subjects without ACL injury, Khanna et al. [20] reported an ALL detection rate of 90%, with the proximal femoral attachment site being unclear. Similarly, Klontzas et al. [18] reported a comparable detection rate but noted unclarity at the distal tibial attachment site of the ALL. While the ALL detection rate in the group without ACL injury was similar to our results, existing studies primarily described the presence, path, and dimensions of the ALL rather than its detailed morphology. In contrast, we classified the visibility of the ALL based on its morphology, distinguishing between normal, probably normal, abnormal, and invisible. The ALL was considered normal if it appeared straight, had a consistent thickness, and was thicker than or equal to the MTL. A probably normal classification was given when the ALL exhibited a continuous, wavy appearance and was thinner than the MTL. Approximately 63.7–71.2% of patients belonged to the normal or probably normal group, displaying continuous ligamentous features. Conversely, 19.7–30.3% exhibited abnormal features, such as discontinuity or angulation. A small percentage (4.5–12.1%) of cases fell into the invisible group, as no distinct ligamentous structure was observed.

The reported dimensions of ALL were 4 to 7 mm in width and 1 to 2 mm in thickness. [9,18,29]. This study aimed to evaluate the thickness of the ALL by comparing it with the thickness of the MTL. Since the MTL structure is always present, it provides an easy and intuitive reference for assessing the thickness of the ALL. Among all the patients, approximately 36.4% had an ALL thickness equal to or greater than the MTL, while 27.3% exhibited a thinner but continuous pattern compared to the MTL. Therefore, the most common characteristic of the ALL in young adults without ACL injury was the presence of continuous ligaments that were equal to or thicker than the MTL. It is worth noting that the evaluation of width was not possible in this study due to the use of coronal and oblique coronal sequences. Similarly to other studies [18,30], our research recognizes the limitations of fully tracing the path of the ALL on axial MR images.

The femoral origins of the ALL have been described differently in various studies. By using the origin of the FCL as a reference point, the femoral attachments of the ALL were categorized as either posterior and proximal to, parallel to, or anterior and distal to the FCL [13,29]. There is still disagreement regarding the precise location and frequency of the femoral origin. In our study, we did not specify a single point of femoral attachment, and most of the femoral origins were found to be anterior to the FCL origin. Caterine et al. also encountered similar challenges in distinguishing the femoral origin using MRI [21]. Dodds et al. [14] demonstrated that the femoral origin is complex, consisting of a fan-shaped arrangement of fibers without a clear area of direct bony attachment.

However, there is a general consensus regarding the location of the tibial insertion. It is commonly accepted that it is situated at the midpoint between Gerdy’s tubercle and the tip of the fibular head, approximately 4.0 to 7.0 mm below the tibial plateau [31,32]. In our study, we classified the distal insertion of the ALL based on its relationship to the tibial insertion of the MTL. In our study, we found that the tibial attachment was equally present either at the same insertion or distal to the MTL. On average, the distance from the distal attachment of the ALL to the MTL attachment was 3.7 mm (range, 1.7–6.1 mm). Differences in measurements may have been influenced by the fact that the insertion sites of MTL and ALL may not be in the same coronal plane. To measure the perpendicular distance between the insertion points of MTL and ALL, we used axial T2-weighted fat-suppressed images.

Several studies have reported that the ALL has two distal insertion sites [2,12,33]. One is in the tibial plateau and the other is in the lateral meniscus, but there is still considerable debate about the latter [14,34]. No meniscal insertion was found in the current study. Our findings align with Taneja et al.‘s study [17], which revealed that only the distal tibial insertion site of the ALL was identifiable on MRI, while the meniscal insertion site was not discernible in all observed cases. It is possible that the meniscal insertion site is located within the meniscofemoral ligament or could be too thin or unclear to be detected on MRI.

In addition to our main findings, we also identified other characteristics of ALL in our study. Ligamentous duplication was observed in six cases (9.1%), while intrasubstance splitting was found in two cases (3.0%). It is worth noting that all the duplicated or split cases showed similar attachment sites at the proximal and distal ends. These findings align with a previous study conducted by Helito et al. [35], who also reported the presence of multiple bands in the structure of the ALL during anatomical dissection. However, the incidence of our study’s results differs from the study that reported similar findings in approximately 92.3% of cases. This discrepancy is likely attributed to the proximity and intermingling of the small anatomical components within the lateral knee ligament complex. This can lead to misidentification as part of the ALL during dissection or imaging evaluation.

Our study had several limitations. Firstly, it was designed retrospectively, relying on departmental protocols without thin-sliced, volumetric imaging sequences. We did not include fat-suppressed 3D-isotropic images for image analysis because blurring and low contrast-to-noise ratio make it difficult to distinguish the detailed anatomy of thin and small structures. A direct correlation with anatomical dissections was not available since the ALL is an extra-articular structure and cannot be examined directly using arthroscopy. We acknowledge the inherent limitations of MRI resolution in evaluating small ligamentous structures, which may lead to misinterpretations as true anatomical structures or pseudolesions. Secondly, it is worth mentioning that this study had a small sample size. However, it is important to note that there is a lack of studies on the imaging characteristics of ALL in patients without ACL injury. While there is existing research on the ALL using MRI, previous studies have mainly focused on the ALL in the context of ACL injury. In contrast, our study included patients with presumably intact ALL but without ACL injury. Given that this study is a preliminary evaluation conducted by two radiologists, further validation with additional data will be necessary. Thirdly, it is important to consider that the chronicity of ligament injury was not taken into account in our MRI analysis. Even though the ligaments appeared intact, previously undetected tears may have caused the ligament to become thick or thin. Despite these limitations, we believe that our findings are representative of imaging routines found in most imaging services and show good agreement with the expertise of independent readings by two experienced radiologists. Understanding the MRI features of an intact ALL without ACL injury may improve the knowledge of its potential role, help identify the pathology associated with ALL, and contribute to personalized treatment approaches, especially in ACL reconstruction. Our study may serve as an anatomical guide for interpreting conventional MRI in the context of potential anterolateral knee injuries and provide a basis for future clinical studies correlating ALL findings with clinical signs of instability. Additionally, a comparative study of the MRI characteristics of the ALL with and without ACL injury may be worth considering as a future direction for research.

In conclusion, MRI was found to be a feasible method for detecting the ALL in the majority of young adults without ACL injury. MRI scans showed continuous ligamentous structures in approximately two-thirds of the cases, with around 40% exhibiting thicknesses equal to or greater than the MTL. The majority of the ligament’s proximal attachments were located anterior to the FCL, while the distal attachments were equally distributed between the same insertion site as the MTL or distal to it.

## Figures and Tables

**Figure 1 diagnostics-14-01226-f001:**
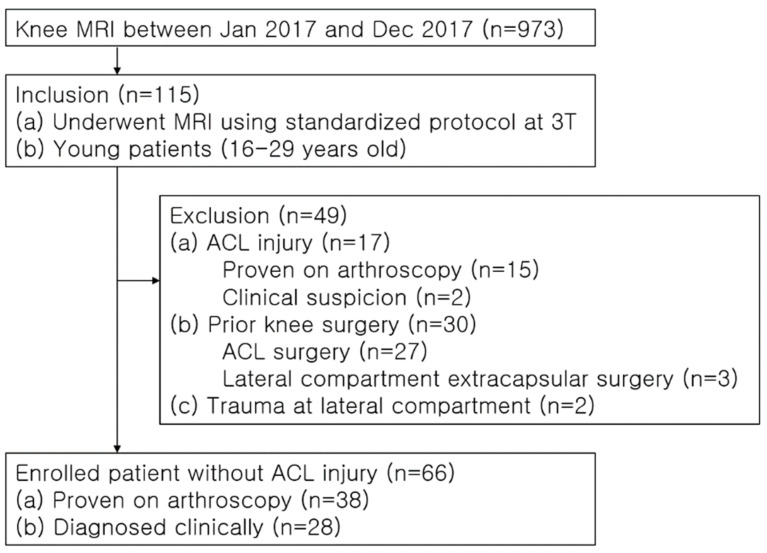
Flow diagram of the study population.

**Figure 2 diagnostics-14-01226-f002:**
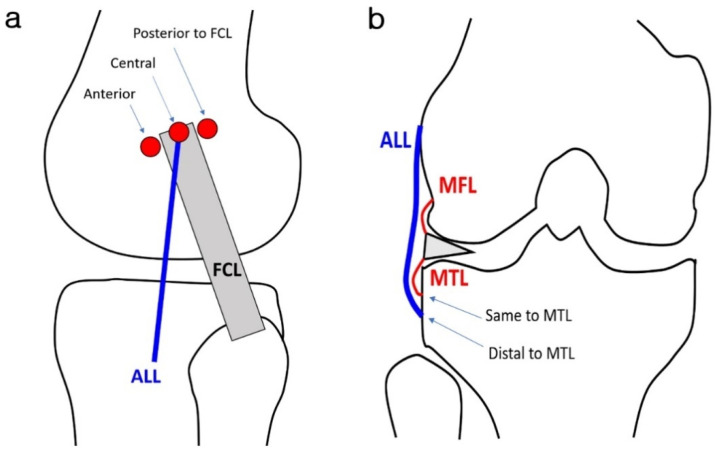
Schematic anatomical illustrations of the anterolateral ligament. The ALL attaches from the lateral epicondyle of the femur to the lateral tibia. (**a**) In the sagittal plane, the proximal femoral attachment site of the ALL was classified as anterior, center, posterior to the FCL. (**b**) In the coronal plane, the distal tibial attachment site was classified as the same insertion, distal insertion to the MTL. ALL, anterolateral ligament; MFL, medial meniscofemoral capsular ligament; MTL, medial meniscotibial ligament; FCL, fibular collateral ligament.

**Figure 3 diagnostics-14-01226-f003:**
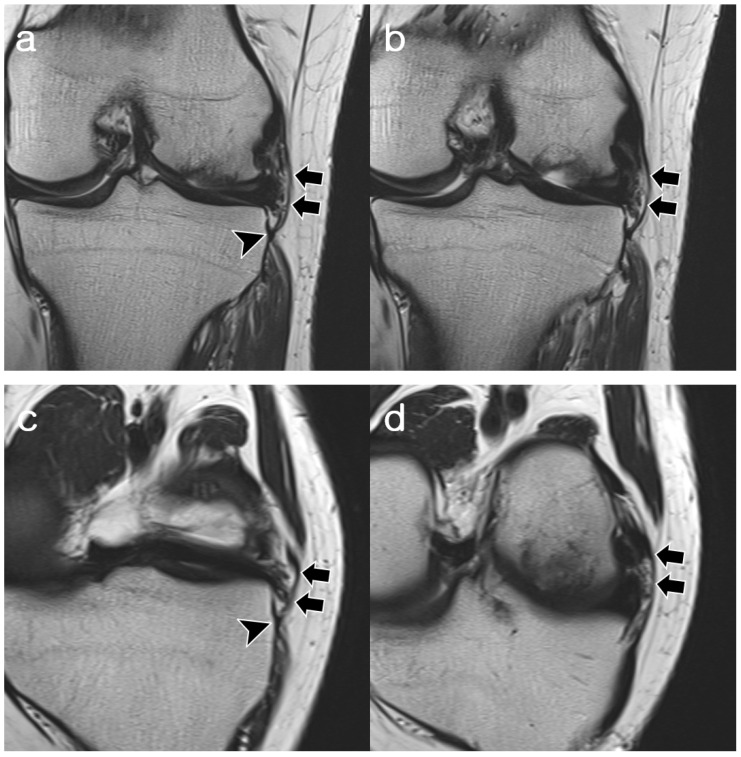
A 22-year-old woman with osteochondritis dissecans of the lateral femoral condyle. (**a**,**b**) Coronal T2-weighted images and (**c**,**d**) oblique coronal T2-weighted images show fully visible, convex, continuous ligamentous structures of the ALL (arrows), which are uniform and of equal thickness to the MTL (arrowheads). The proximal attachment of the ALL is located anterior to the origin of the FCL. Distal attachment is identical to MTL. Both readers assigned the ALL to “normal” appearance during both reading sessions. ALL, anterolateral ligament; MTL, medial meniscotibial ligament; FCL, fibular collateral ligament.

**Figure 4 diagnostics-14-01226-f004:**
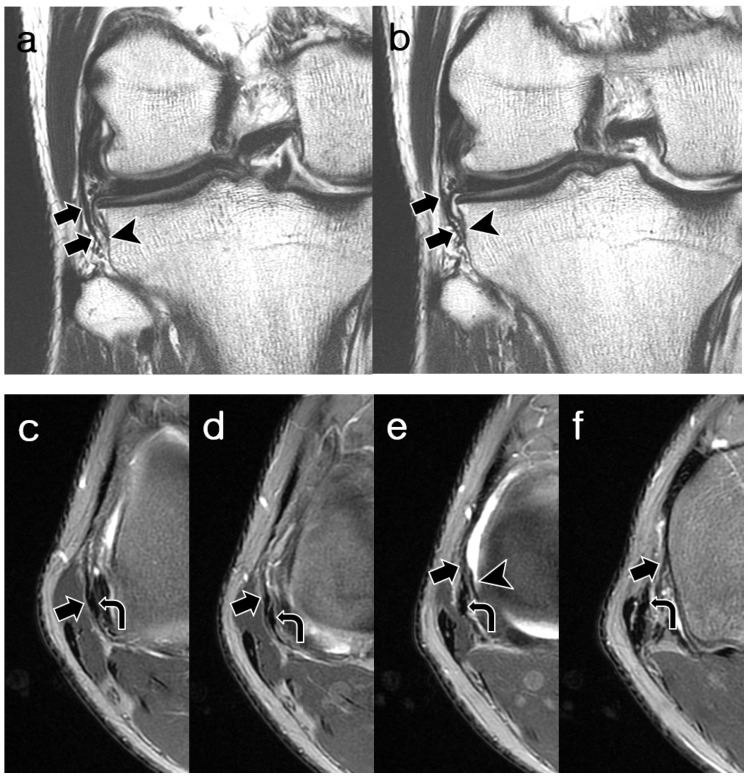
A 21-year-old man with discoid lateral meniscus and a history of medial meniscectomy. (**a**,**b**) Coronal T2-weighted images and (**c**–**f**) axial T2-weighted images with fat suppression show a fully visible, convex, and continuous structure of ALL (arrows), which is slightly thicker than the MTL (arrowheads). The proximal attachment of the ALL is centered to the origin of the FCL (curved arrows). Distal attachment is located 5.1 mm distal to the MTL. The ALL was assigned “normal” appearance by both readers. However, one reader assigned it “probably normal” due to uneven thickness in one session. ALL, anterolateral ligament; MTL, medial meniscotibial ligament; FCL, fibular collateral ligament.

**Figure 5 diagnostics-14-01226-f005:**
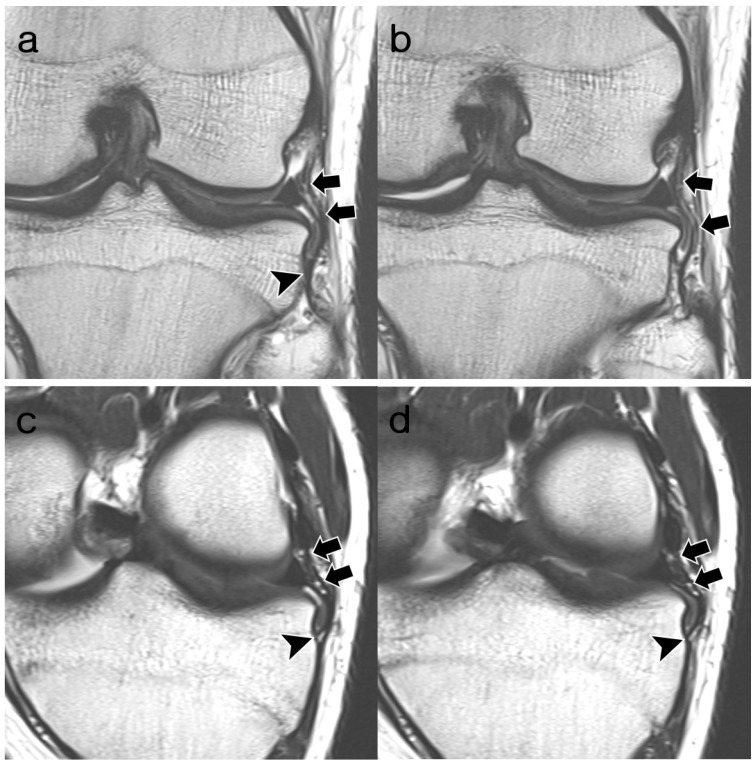
A 17-year-old man with cartilage erosion of the patella. (**a**,**b**) Coronal T2-weighted images and (**c**,**d**) oblique coronal T2-weighted images show a fully visible and continuous but wavy contoured structure of ALL (arrows), which is slightly thinner than the MTL (arrowheads). The proximal attachment of the ALL is centered to the origin of the FCL. Distal attachment is identical to MTL. The ALL was assigned “probably normal” appearance by both readers. ALL, anterolateral ligament; MTL, medial meniscotibial ligament; FCL, fibular collateral ligament.

**Figure 6 diagnostics-14-01226-f006:**
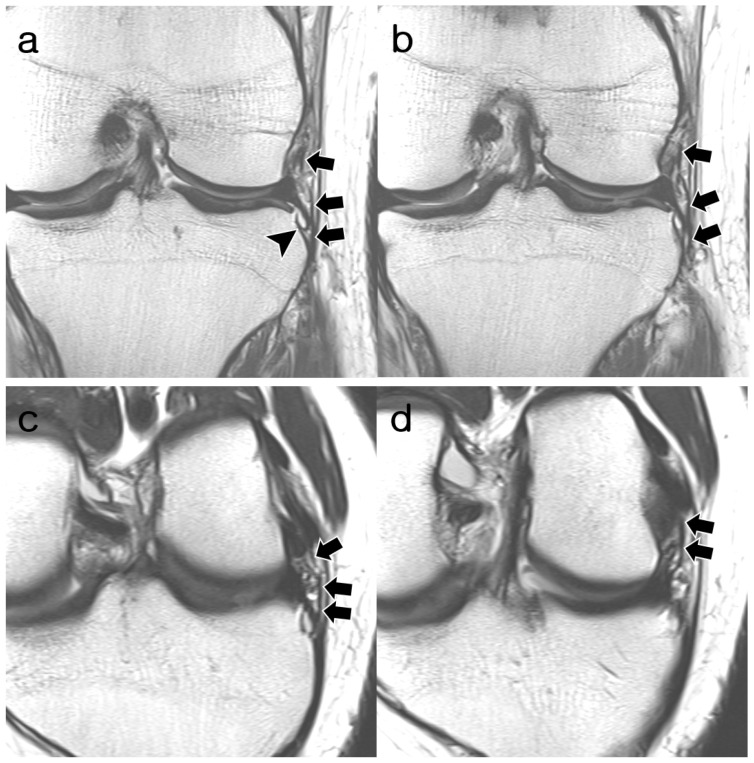
A 23-year-old man with cartilage erosion of medial femoral condyle. (**a**,**b**) Coronal T2-weighted images and (**c**,**d**) oblique coronal T2-weighted images show an angulation and discontinuity at the mid-substance of ALL (arrows), which is slightly thinner than the MTL (arrowheads). The proximal attachment of the ALL is located anterior to the origin of the FCL. Distal attachment is located 5.5 mm distal to the MTL. Both readers assigned ALL to “abnormal” appearance during each session. ALL, anterolateral ligament; MTL, medial meniscotibial ligament; FCL, fibular collateral ligament.

**Figure 7 diagnostics-14-01226-f007:**
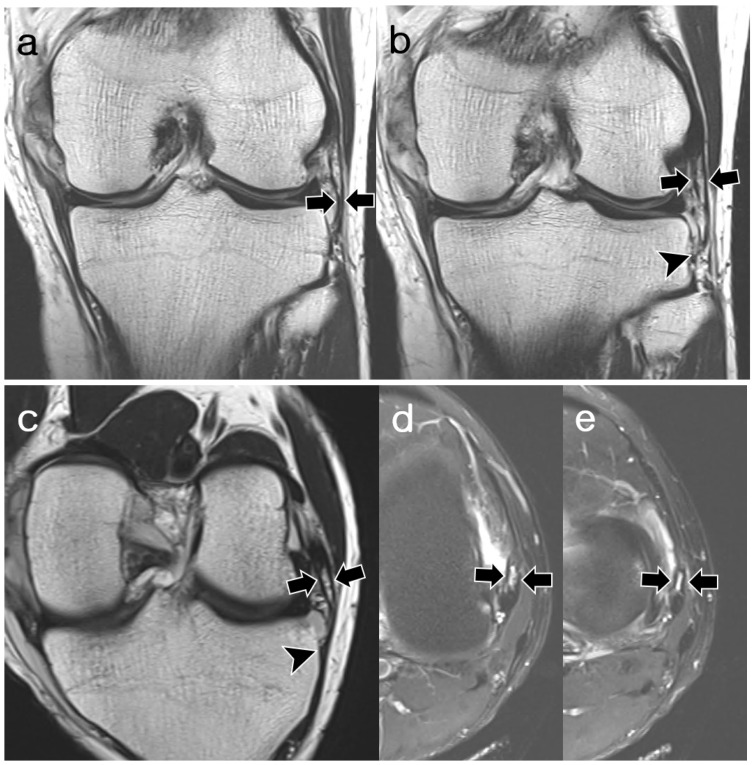
A 29-year-old male with complete medial collateral ligament tear. (**a**,**b**) Coronal T2-weighted images, (**c**) oblique coronal T2-weighted images, and (**d**,**e**) axial T2-weighted images with fat suppression demonstrate fully visible, straight, and continuous structures of double ALL (arrows). Both are slightly thicker than the MTL (arrowheads). The proximal attachments of the ALL are located superficial and deep to the center of the FCL origin. The two parts of ALL were joined distally and attached to the same area as the MTL. Both readers assigned ALL to “normal” appearance during each session. ALL, anterolateral ligament; MTL, medial meniscotibial ligament; FCL, fibular collateral ligament.

**Table 1 diagnostics-14-01226-t001:** MRI parameters.

Parameters	Magnetom Skyra			Discovery MR750		
	Cor-T2	Ax-T2-FS	Obl-cor-T2	Cor-T2	Ax-T2-FS	Obl-cor-T2
TR/TE (ms)	3900/76	4130/58	3400/68	3750/73	3500/64	2500/71
Flip angle (°)	150	150	150	142	142	142
Matrix size	512 × 281	384 × 269	384 × 211	640 × 320	352 × 288	416 × 256
Field of view (cm)	16	15	14	16	15	14
Section thickness (mm)	3	4	3	3	4	3
Intersection gap (mm)	0	0	0	0	0	0
Bandwidth (kHz/pixel)	200	205	205	98	98	195
Echo train length	11	13	15	10	14	12
Number of excitation	2	3	4	1	2	2

Cor-T2 = coronal T2-weighted sequence, Ax-T2-FS = axial T2-weighted sequence with fat suppression, Obl-cor-T2 = oblique coronal T2-weighted sequence, TR = repetition time, TE = echo time.

**Table 2 diagnostics-14-01226-t002:** Visibility grade of anterolateral ligament.

	Session 1		Session 2	
	Reader 1	Reader 2	Reader 1	Reader 2
Normal	24 (36.4)	20 (30.3)	25 (37.9)	24 (36.4)
Probably normal	20 (30.3)	27 (40.9)	18 (27.3)	18 (27.3)
abnormal	19 (28.8)	13 (19.7)	20 (30.3)	16 (24.2)
Non-visualized	3 (4.5)	6 (9.1)	3 (4.5)	8 (12.1)

Data are presented as the number of the knees (percentage).

**Table 3 diagnostics-14-01226-t003:** Proximal and distal attachment of anterolateral ligament.

Proximal attachment of ALL	
Anterior to FCL	37 (63.8)
Central to FCL	15 (25.9)
Posterior to FCL	1 (1.7)
Mid of FCL	4 (6.9)
Blended into meniscofemoral ligament	1(1.7)
Distal attachment of ALL	
Same insertion as MTL	29 (50)
Distal insertion to MTL	29 (50)

Data are presented as the number of the knees (percentage). ALL, anterolateral ligament; FCL, fibular collateral ligament; MTL, medial meniscotibial ligament.

## Data Availability

The data presented in this study are available upon request from the corresponding author.

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
