# Peer review of "Magnetic Resonance Imaging Features of Anterolateral Ligament in Young Adults without Anterior Cruciate Ligament Injury: Preliminary Evaluation"

_diagnostics, 2024, doi:10.3390/diagnostics14121226_

Round 1

Reviewer 1 Report

Comments and Suggestions for Authors

Title: do not use abbreviations in the title

Abstract

any hypothesis?

report inclusion and exclusion criteria

introduction

little bit too long. please focus on the rationale of your study

report what is known

analyze controversies

finish with aim and hypotehsis

methods

strobe checklist are missing

ethical approval is missing

report clear inclusion and exclusion criteria

was a power analysis performed to ensure sample size

results

well written but little bit too long

try to reduce

discussion

start with main findings of your results

report what is new

analyze similar studies in literature

discuss about the clinical impact of your study

improve limitations section

conclusions

coherent

references: add following

D'Ambrosi R, Sconfienza LM, Albano D, Messina C, Mangiavini L, Ursino N, Rinaldi S, Zanirato A, Tagliafico A, Formica M. High incidence of RAMP lesions and a nonnegligible incidence of anterolateral ligament and posterior oblique ligament rupture in acute ACL injury. Knee Surg Sports Traumatol Arthrosc. 2024 Apr 30. doi: 10.1002/ksa.12219. Epub ahead of print. PMID: 38686571.

Mogos Ș, Antonescu D, Stoica IC, D'Ambrosi R. Superior rotational stability and lower re-ruptures rate after combined anterolateral and anterior cruciate ligament reconstruction compared to isolated anterior cruciate ligament reconstruction: a 2-year prospective randomized clinical trial. Phys Sportsmed. 2023 Aug;51(4):371-378. doi: 10.1080/00913847.2022.2112914. Epub 2022 Aug 18. PMID: 35968574.

Comments on the Quality of English Language

good

Author Response

Authors' Responses to Reviewer's Comments (Reviewer 1)

Author's Notes

Reviewer 1, Comment 1. Title: do not use abbreviations in the title

-> Thank you for your insightful comment. As you requested, we have modified the title as follows.

Magnetic resonance imaging features of anterolateral ligament in young adults without anterior cruciate ligament injury: preliminary evaluation

Abstract

Reviewer 1, Comment 2. any hypothesis?

-> Thank you for your suggestion. We have added our hypothesis in introduction section as requested.

We hypothesized that in subjects without ACL injury, where the ALL is expected to be intact, the ALL would have unique appearance on MRI and distinct proximal and distal attachment sites compared to other fascia or retinaculum structures.

Reviewer 1, Comment 3. report inclusion and exclusion criteria

-> Thank you for your kind comment. As you requested, we have clarified the inclusion and exclusion criteria and provided a flow chart in Materials and Methods section.

Between January and December 2017, our hospital conducted 973 knee MRI scans. Of these, 115 met the inclusion criteria: (a) 3-T MRI study with a standardized protocol and (b) patients aged 16 to 29 years. We excluded 49 scans due to prior ACL or lateral compartment capsule surgery (30 cases), confirmed ACL injuries via arthroscopy (15 cases), clinical diagnoses of ACL injury with pivot-shift bone marrow edema (2 cases), and lateral compartment fractures (2 cases). Consequently, 66 knee MRI scans were included in the study (Figure 1). Electronic medical records and surgical notes were reviewed for each case, documenting age, gender, surgical history, and follow-up period.

 introduction

Reviewer 1, Comment 4. little bit too long. please focus on the rationale of your study

-> Thank you for your constructive feedback. As you pointed out, we have shortened the introduction, focusing on the rationale of our study.

Reviewer 1, Comment 5. report what is known

-> We appreciate your suggestion. As requested, we have included current known anatomical and MR imaging understandings of ALL with respect to its respective features such as presence, path and attachments. Please consider this.

Reviewer 1, Comment 6. analyze controversies

-> Thank you for your kind comment. As requested, we have analyzed the controversies.

Previous studies have employed MRI to assess the ALL primarily in cadaver knees or in the context of ACL injuries. However, there is a notable lack in research on the MRI characteristics of the ALL in the absence of ACL injury [18,20].

Reviewer 1, Comment 7. finish with aim and hypothesis

-> Thank you for your kind comment. As you requested, we have finished the introduction section with our hypothesis and aim.

We hypothesized that in subjects without ACL injury, where the ALL is expected to be intact, the ALL would have unique appearance on MRI and distinct proximal and distal attachment sites compared to other fascia or retinaculum structures. The purpose of this study is to describe the MRI imaging characteristics of ALL in young adults without ACL injury, focusing on its morphology, continuity, and attachment points. Additionally, we aim to assess the visibility of the ALL using routine 3-Tesla MRI.

methods

Reviewer 1, Comment 8. strobe checklist are missing

-> Thank you for your valuable comment. We have ensured our study complies with the STROBE guideline. As you pointed out, we have added a sentence explaining this.

This study was conducted following the STROBE (Strengthening the Reporting of Observational Studies in Epidemiology) guidelines and complied with the ethical standards established by the 1964 Declaration of Helsinki and its subsequent amendments.

Reviewer 1, Comment 9. ethical approval is missing

-> Thank you for your kind comment. As requested, we have added a sentence regarding ethical approval.

This retrospective study received approval from our hospital's institutional review board, which waived the need for informed consent due to the study's retrospective design.

Reviewer 1, Comment 10. report clear inclusion and exclusion criteria

-> Thank you for your kind comment. As mentioned in response to comment 3, we have clarified the inclusion and exclusion criteria and provided a flow chart in Materials and Methods section. Please consider this.

Reviewer 1, Comment 11. was a power analysis performed to ensure sample size

-> Thank you for your suggestion. As requested, we have included a paragraph detailing the power analysis to ensure the sample size was adequate.

Power analysis for the paired sample t-tests, setting an alpha error at 0.05 and aiming for a power of 0.8, determined that a minimum sample size of 29 knees was necessary. Power analysis was conducted using G*Power software 3.1.9.4 (Heinrich-Heine-Universität Düsseldorf, Düsseldorf, Germany).

results

Reviewer 1, Comment 12. well written but little bit too long, try to reduce

-> Thank you for your kind feedback. As you requested, we have condensed the paragraphs in results section. Please consider this.

discussion

Reviewer 1, Comment 13. start with main findings of your results

-> Thank you for your kind comment. As you requested, we have started the discussion section with main findings of our results.

The main results of this study show that in approximately 90% of young adults without ACL injury, ALL was detectable on MRI. In about two-thirds of the cases, MRI revealed continuous ligament structures. Additionally, around 40% of these structures were found to be either thicker than or similar in size to the MTL. The majority of the proximal attachment was found to be located anterior to the FCL, while the distal attachment was found to be either at the same insertion point or distal to the MTL with equal frequency.

Reviewer 1, Comment 14. report what is new

-> Thank you for your suggestion. As you requested, we have reported the new findings of our study.

Approximately 63.7-71.2% of patients belonged to the normal or probably normal group, displaying continuous ligamentous features. Conversely, 19.7-30.3% exhibited abnormal features, such as discontinuity or angulation. A small percentage (4.5-12.1%) of cases fell into the invisible group, as no distinct ligamentous structure was observed.

Therefore, the most common characteristic of the ALL in young adults without ACL injury was the presence of continuous ligaments that were equal to or thicker than the MTL.

Reviewer 1, Comment 15. analyze similar studies in literature

-> Thank you for your thoughtful comment. As you requested, we have analyzed similar studies in the literature.

The majority of existing studies have focused on evaluating the ALL in cadaveric knees or in the context of ACL injuries. There is notable paucity of research examining the MRI characteristics of the ALL in the absence of ACL injury [18,20]. In studies involving subjects without ACL injury, Khanna et al. [20] reported an ALL detection rate of 90%, with the proximal femoral attachment site being unclear. Similarly, Klontzas et al. [18] reported a comparable detection rate but noted unclarity at the distal tibial attachment site of the ALL. While the ALL detection rate in the group without ACL injury was similar to our results, existing studies primarily described the presence, path, and dimensions of the ALL rather than its detailed morphology. In contrast, we classified the visibility of the ALL based on its morphology, distinguishing between normal, probably normal, abnormal, and invisible.

Reviewer 1, Comment 16. discuss about the clinical impact of your study

-> Thank you for your insightful comment. As you requested, we have discussed the clinical impact of our study.

Despite these limitations, we believe that our findings are representative of imaging routines found in most imaging services and show good agreement with the expertise of independent readings by two experienced radiologists. Understanding the MRI features of intact ALL without ACL injury may improve knowledge of its potential role, help identify pathology associated with ALL, and contribute to personalized treatment approaches, especially in ACL reconstruction. Our study may serve as an anatomical guide for interpreting conventional MRI in the context of potential anterolateral knee injuries and provide a basis for future clinical studies correlating ALL findings with clinical signs of instability.

Reviewer 1, Comment 17. improve limitations section

-> Thank you for your suggestion. As you requested, we have revised and enhanced the limitations section.

conclusions

coherent

references:

Reviewer 1, Comment 18. add following

 D'Ambrosi R, Sconfienza LM, Albano D, Messina C, Mangiavini L, Ursino N, Rinaldi S, Zanirato A, Tagliafico A, Formica M. High incidence of RAMP lesions and a nonnegligible incidence of anterolateral ligament and posterior oblique ligament rupture in acute ACL injury. Knee Surg Sports Traumatol Arthrosc. 2024 Apr 30. doi: 10.1002/ksa.12219. Epub ahead of print. PMID: 38686571.

 Mogos Ș, Antonescu D, Stoica IC, D'Ambrosi R. Superior rotational stability and lower re-ruptures rate after combined anterolateral and anterior cruciate ligament reconstruction compared to isolated anterior cruciate ligament reconstruction: a 2-year prospective randomized clinical trial. Phys Sportsmed. 2023 Aug;51(4):371-378. doi: 10.1080/00913847.2022.2112914. Epub 2022 Aug 18. PMID: 35968574.

-> Thank you for your suggestion. As requested, we have added the reference notations and related content in discussion section.

ALL injuries are commonly associated with ACL injuries, with MRI reports showing prevalence rates as high as 79% in ACL injured knees [22]. Recent reports indicate that 19.2% to 29.3% of patients with a complete ACL tear also have a concomitant ALL tear on MRI [23,24]. Many institutions have adopted additional ALL reconstructions for ACL injuries with rotational instability. Combined ACL and ALL reconstruction has demonstrated greater mid-term rotational stability and significantly lower re-rupture rates compared to isolated ACL reconstruction [25].

Reviewer 2 Report

Comments and Suggestions for Authors

Thank you for the opportunity to review this very interesting manuscript.

This work presents some interesting conclusions. However, before it is published, the following points should be revised:

1-   The title of the manuscript should be revised, I don't think it should start with acronyms and, since it is an evaluation carried out by two readers, it should indicate that it is a pilot study

2- Always throughout the text when you write an acronym for the first time you must write the capital letter in the text that refers to the acronym (I have identified some notes in the text)

3 – When identify some authors with their name and et al. the name al. needs a full stop. Sometimes you put the dot, sometimes you don't. You should check it again.

4 – Not start a sentence with a number

5 - For publication the table 1 should be on the same page

6 - The readers' experience will influence the results, what do you think? Even if you choose readers with the same experience, the results may be different, because it will be a qualitative analysis and not a quantified one. We should have data from more readers to validate the results. However, as a preliminary approach, it seems to me that the manuscript should be published in order to improve the analysis carried out.

7 - Even if you choose readers with the same experience, the results may be different, because it will be a qualitative analysis and not a quantified one. We should have data from more readers to validate the results.

8- When write the name of the software, the name needs Trade Mark ™.

Congratulations on your work.

Author Response

Authors' Responses to Reviewer's Comments (Reviewer 2)

Author's Notes

Thank you for the opportunity to review this very interesting manuscript.

This work presents some interesting conclusions. However, before it is published, the following points should be revised:

Reviewer 2, Comment 1.   The title of the manuscript should be revised, I don't think it should start with acronyms and, since it is an evaluation carried out by two readers, it should indicate that it is a pilot study

-> Thank you for your insightful comment. We agree with your observation. As suggested, we have revised the title as follows.

Magnetic resonance imaging features of anterolateral ligament in young adults without anterior cruciate ligament injury: preliminary evaluation

Reviewer 2, Comment 2. Always throughout the text when you write an acronym for the first time you must write the capital letter in the text that refers to the acronym (I have identified some notes in the text)

-> Thank you for your detailed comment. As you indicated, we have corrected all acronyms throughout the text.

Reviewer 2, Comment 3. When identify some authors with their name and et al. the name al. needs a full stop. Sometimes you put the dot, sometimes you don't. You should check it again.

-> Thank you for your kind comment. As you indicated, we have reviewed and corrected the format of authors’ names throughout the text consistently.

Reviewer 2, Comment 4. Not start a sentence with a number

-> Thank you for your valuable comment. As you pointed out, we have revised sentences to avoid starting with numbers throughout the text.

Reviewer 2, Comment 5. For publication the table 1 should be on the same page

-> Thank you for your practical comment. As you pointed out, we have rearranged the table to be displayed on a single page.

Reviewer 2, Comment 6. The readers' experience will influence the results, what do you think? Even if you choose readers with the same experience, the results may be different, because it will be a qualitative analysis and not a quantified one. We should have data from more readers to validate the results. However, as a preliminary approach, it seems to me that the manuscript should be published in order to improve the analysis carried out.

-> Thank you for your insightful comment. We agree with your assessment. We have addressed this in limitation session and modified the title to reflect the preliminary nature.

Given that this study is a preliminary evaluation conducted by two radiologists, further validation with additional data will be necessary.

Reviewer 2, Comment 7. When write the name of the software, the name needs Trade Mark ™.

-> Thank you for your detailed comment. As you indicated, we have corrected the software name to include the trade mark symbol.

Congratulations on your work.

Reviewer 3 Report

Comments and Suggestions for Authors

The study provides a detailed analysis of the MRI appearance of the anterolateral ligament (ALL) in young adults without anterior cruciate ligament (ACL) injury. It describes the MRI characteristics (pathway, proximal and distal attachments) that can be used to identify the intact ALL in individuals without ACL injury. It also discusses how MRI can be used to visualise and evaluate the ALL when it is expected to be intact, before any potential damage from ACL injury.

Some recommendations to improve the manuscript:

1. The authors should state the gap in the existing literature, particularly emphasising the novelty of studying the ALL in individuals without ACL injury in the introduction.

2. Consider reframing the study objectives to make them more specific. For example, rather than broadly assessing the MRI features, specify which features (e.g., visibility, continuity, attachment points) are being scrutinised.

3. Consider adding a comparative group, such as individuals with ACL injuries, to highlight differences in ALL visibility or morphology.

4. The authors may consider deepening the discussion around why certain findings (e.g., the visibility rates or morphological features) might differ from previous studies. This could involve more comprehensive comparisons with existing literature.

5. It would be great if the authors could address the clinical implications of your findings more thoroughly, discussing how this might influence clinical practice or future research directions.

6. Perhaps expand the discussion to compare the findings here to other studies that have looked at the normal anatomy and MRI appearance of the ALL. Highlight similarities and differences.

7. More critically discuss the study's limitations. For instance, acknowledge the limitations of MRI resolution in evaluating small ligamentous structures and how this might affect the findings.

Comments on the Quality of English Language

Minor editing is required.

Author Response

Authors' Responses to Reviewer's Comments (Reviewer 3)

Author's Notes

The study provides a detailed analysis of the MRI appearance of the anterolateral ligament (ALL) in young adults without anterior cruciate ligament (ACL) injury. It describes the MRI characteristics (pathway, proximal and distal attachments) that can be used to identify the intact ALL in individuals without ACL injury. It also discusses how MRI can be used to visualise and evaluate the ALL when it is expected to be intact, before any potential damage from ACL injury.

Some recommendations to improve the manuscript:

Reviewer 3, Comment 1. The authors should state the gap in the existing literature, particularly emphasising the novelty of studying the ALL in individuals without ACL injury in the introduction.

-> Thank you for your insightful comment. The lack of research on the MRI characteristics of the ALL in the absence of ACL injury is noticeable. (In our text, 'Gap' was a typographical error, and 'lack' was intended term.) We have underscored this deficiency in the existing literature by highlighting the paucity of research in both the introduction and discussion sections.

Previous studies have employed MRI to assess the ALL primarily in cadaver knees or in the context of ACL injuries. However, there is a notable lack in research on the MRI characteristics of the ALL in the absence of ACL injury [18,20].

The majority of existing studies have focused on evaluating the ALL in cadaveric knees or in the context of ACL injuries. There is notable paucity of research examining the MRI characteristics of the ALL in the absence of ACL injury [18,20]. In studies involving subjects without ACL injury, Khanna et al. [20] reported an ALL detection rate of 90%, with the proximal femoral attachment site being unclear. Similarly, Klontzas et al. [18] reported a comparable detection rate but noted unclarity at the distal tibial attachment site of the ALL. While the ALL detection rate in the group without ACL injury was similar to our results, existing studies primarily described the presence, path, and dimensions of the ALL rather than its detailed morphology. In contrast, we classified the visibility of the ALL based on its morphology, distinguishing between normal, probably normal, abnormal, and invisible.

Reviewer 3, Comment 2. Consider reframing the study objectives to make them more specific. For example, rather than broadly assessing the MRI features, specify which features (e.g., visibility, continuity, attachment points) are being scrutinised.

 -> Thank you for your valuable suggestion. As you requested, we have modified the study objectives to be more specific as follows.

The purpose of this study is to describe the MRI imaging characteristics of ALL in young adults without ACL injury, focusing on its morphology, continuity, and attachment points. Additionally, we aim to assess the visibility of the ALL using routine 3-Tesla MRI.

Reviewer 3, Comment 3. Consider adding a comparative group, such as individuals with ACL injuries, to highlight differences in ALL visibility or morphology.

-> Thank you for your thoughtful comment. This study is a preliminary evaluation to investigate the MRI characteristics of the ALL in a group of young patients without ACL injury. Therefore, comparing ALL characteristics in patients with ACL injury is considered out of scope. However, we acknowledge the value of such a comparative study and have mentioned it as a potential future research direction in the limitation session.

Our study may serve as an anatomical guide for interpreting conventional MRI in the context of potential anterolateral knee injuries and provide a basis for future clinical studies correlating ALL findings with clinical signs of instability. Additionally, a comparative study of MRI characteristics of ALL with and without ACL injury may be worth considering as a future research.

Reviewer 3, Comment 4. The authors may consider deepening the discussion around why certain findings (e.g., the visibility rates or morphological features) might differ from previous studies. This could involve more comprehensive comparisons with existing literature.

-> Thank you for your kind comment. As you requested, we have expanded the discussion to include more comprehensive comparisons with existing literature, addressing why certain findings might differ.

Ariel et al. [19], in a systematic review of anatomical studies, reported that the ALL was observed in 84.8% of overall MRI studies. However, the detection rates in dissection studies varied significantly, ranging from 4% to 100% [9,27]. These discrepancies between MRI and cadaveric dissection detection rates may be attributed to technical variations. Evaluating the anatomical structure of the ALL based solely on coronal MRI images, as conducted by Claes et al. [22] and Helito et al. [28], may lead to potential anatomical pitfalls and partial-volume effects, particularly given the ALL's thin, short structure and its close intertwinement with the surrounding fibers of the lateral ligamentous complex of the knee [14]. In contrast, our study employed specific criteria for assessing the visibility of the ALL using both coronal and oblique coronal sequences, resulting in MRI detection rate for the ALL between 87.9% and 95.5%.

Reviewer 3, Comment 5. It would be great if the authors could address the clinical implications of your findings more thoroughly, discussing how this might influence clinical practice or future research directions.

-> Thank you for your insightful comment. As you requested, we have added a paragraph discussing the clinical implications of our findings more thoroughly in the discussion section.

Despite these limitations, we believe that our findings are representative of imaging routines found in most imaging services and show good agreement with the expertise of independent readings by two experienced radiologists. Understanding the MRI features of intact ALL without ACL injury may improve knowledge of its potential role, help identify pathology associated with ALL, and contribute to personalized treatment approaches, especially in ACL reconstruction. Our study may serve as an anatomical guide for interpreting conventional MRI in the context of potential anterolateral knee injuries and provide a basis for future clinical studies correlating ALL findings with clinical signs of instability. Additionally, a comparative study of MRI characteristics of ALL with and without ACL injury may be worth considering as a future research.

Reviewer 3, Comment 6. Perhaps expand the discussion to compare the findings here to other studies that have looked at the normal anatomy and MRI appearance of the ALL. Highlight similarities and differences.

-> Thank you for your valuable comment. As you requested, we have expanded the discussion to compare our findings with other studies that have examined the normal anatomy and MRI appearance of the ALL, highlighting similarities and differences for each ALL feature.

The reported dimensions of ALL were 4 to 7 mm in width and 1 to 2 mm in thickness. [9,18,29]. This study aimed to evaluate the thickness of the ALL by comparing it with the thickness of the MTL. Since the MTL structure is always present, it provides an easy and intuitive reference for assessing the thickness of the ALL. Among all the patients, approximately 36.4% had an ALL thickness equal to or greater than the MTL, while 27.3% exhibited a thinner but continuous pattern compared to the MTL. Therefore, the most common characteristic of the ALL in young adults without ACL injury was the presence of continuous ligaments that were equal to or thicker than the MTL. It's worth noting that the evaluation of width was not possible in this study due to the use of coronal and oblique coronal sequences. Similarly to other studies [18,30], our research recognizes the limitations of fully tracing the path of the ALL on axial MR images.

Reviewer 3, Comment 7. More critically discuss the study's limitations. For instance, acknowledge the limitations of MRI resolution in evaluating small ligamentous structures and how this might affect the findings.

-> Thank you for your thoughtful comment. As you requested, we have added a paragraph discussing the limitations of MRI resolution in evaluating small ligamentous structures in limitation session.

Firstly, it was designed retrospectively, relying on departmental protocols without thin-sliced, volumetric imaging sequences. We did not include fat suppressed 3D-isotropic images for image analysis because blurring and low contrast-to-noise ratio make it difficult to distinguish detailed anatomy of thin and small structures. Direct correlation with anatomical dissections was not available since the ALL is extra-articular structure and cannot be examined directly by arthroscopy. We acknowledge the inherent limitations of MRI resolution in evaluating small ligamentous structures, which may lead to misinterpretations as true anatomical structures or pseudolesions.